# Graphene-like nanoribbons periodically embedded with four- and eight-membered rings

Meizhuang Liu[1], Mengxi Liu[2], Limin She[1], Zeqi Zha[2], Jinliang Pan[2], Shichao Li[2], Tao Li[3], Yangyong He[1], Zeying Cai[1], Jiaobing Wang[3], Yue Zheng[1], Xiaohui Qiu[2] & Dingyong Zhong[1]

Embedding non-hexagonal rings into $sp^2$-hybridized carbon networks is considered a promising strategy to enrich the family of low-dimensional graphenic structures. However, non-hexagonal rings are energetically unstable compared to the hexagonal counterparts, making it challenging to embed non-hexagonal rings into carbon-based nanostructures in a controllable manner. Here, we report an on-surface synthesis of graphene-like nanoribbons with periodically embedded four- and eight-membered rings. The scanning tunnelling microscopy and atomic force microscopy study revealed that four- and eight-membered rings are formed between adjacent perylene backbones with a planar configuration. The non-hexagonal rings as a topological modification markedly change the electronic properties of the nanoribbons. The highest occupied and lowest unoccupied ribbon states are mainly distributed around the eight- and four-membered rings, respectively. The realization of graphene-like nanoribbons comprising non-hexagonal rings demonstrates a controllable route to fabricate non-hexagonal rings in nanoribbons and makes it possible to unveil their unique properties induced by non-hexagonal rings.

[1] School of Physics and State Key Laboratory for Optoelectronic Materials and Technologies, Sun Yat-Sen University, 510275 Guangzhou, China. [2] CAS Key Laboratory of Standardization and Measurement for Nanotechnology, CAS Center for Excellence in Nanoscience, National Center for Nanoscience and Technology, Beijing 100190, China. [3] School of Chemistry, Sun Yat-Sen University, 510275 Guangzhou, China. Correspondence and requests for materials should be addressed to X.Q. (email: xhqiu@nanoctr.cn) or to D.Z. (email: dyzhong@mail.sysu.edu.cn).

Graphene nanoribbons (GNRs) have attracted extensive attention as promising building blocks for nanoelectronics and spintronics[1–3]. To date, a number of strategies have been developed for the preparation of GNRs. Among them, the 'bottom-up' approach based on the on-surface synthesis from predefined precursor molecules has the advantage to precisely control the edge structure and width[4–7]. In this way, the electronic and magnetic properties, such as band gap and spin-polarized edge states, can be readily tuned[8–10]. The electronic properties of GNRs can also be modulated at nanoscale by chemical doping[11–14] and formation of heterojunctions[15,16]. At the same time, decorating non-hexagonal rings into the honeycomb lattice, which is an effective way to tailor the electronic structures and magnetic properties of such low-dimensional carbon-based structures[17–19], has been intensively studied. Line defects composed of octagons and pentagons in graphene can give rise to the localized electronic states that contribute to the metallic character[20]. The Stone–Wales defect, another typical non-hexagonal structure containing two pentagons and two heptagons, can locally change the density of π-electrons and increase the local reactivity, making it possible to attach metal atoms to further modify the electronic structure[21].

Non-hexagonal rings can be spontaneously formed during graphene preparation[20,22]. For example, in chemical vapour deposition prepared graphene, the grain coalescence results in the formation of linear defects containing pentagons, heptagons and octagons along the boundaries. As for the on-surface synthesized GNRs, penta- and heptagon-rings originating from the interribbon cross-dehydrogenative coupling between the zigzag terminal and the armchair edge of GNRs have been observed[23]. Non-hexagonal rings can also be artificially created by electron irradiation[24], which provides the activation energy higher than the threshold to drive the carbon atom ejection and bond rotation, resulting in randomly arranged polygons. However, it is challenging to periodically embed non-hexagonal rings into carbon-based nanostructures in a controllable manner, limiting the study on how non-hexagonal rings affect their properties.

Here, we report the on-surface synthesis and electronic properties of graphene-like nanoribbons with periodically embedded four- and eight-membered rings. The atomic structure and electronic properties have been investigated by non-contact atomic force microscopy (nc-AFM), scanning tunnelling microscopy (STM) and spectroscopy (STS) combined with density functional theory (DFT) calculations.

## Results

**Synthetic strategy for graphene-like nanoribbons.** Graphene-like nanoribbons comprising four- and eight- membered rings were fabricated by surface-assisted dehalogenation and cyclode-hydrogenation of 1,6,7,12-tetrabromo-3,4,9,10-perylene-tetra-carboxylic-dianhydride ($Br_4$-PTCDA) on Au(111) surfaces, as sketched in Fig. 1. $Br_4$-PTCDA molecules were sublimed onto the Au(111) surface at room temperature under ultrahigh-vacuum (UHV) conditions. Most deposited molecules were intact with the twisted structure owing to the repulsion of bromine substituents (Supplementary Fig. 1; Supplementary Note 1). The C–Br bond cleavage took place at a relatively low temperature of 100 °C due to the steric repulsion between the bromine substituents, yielding the flat $PTCDA-Au_2-Br_4$ hybrids (Supplementary Figs 2a,b and 3). Linear gold–organic hybrid polymers were obtained after the cleavage of Au–Br bonds at 220 °C (ref. 25) (Supplementary Fig. 2c). By further heating to 360 °C, Au–C bonds were cleaved and gold atoms were released from the PTCDA-Au polymers (Supplementary Fig. 2d; Supplementary Note 2). Such a temperature is high enough to drive the symmetric cyclodehydrogenation, resulting in the formation of four- and eight-membered rings between adjacent perylene backbones.

**Synthesis and characterization of graphene-like nanoribbons.** Owing to the instability of anhydride groups at high temperatures, side reactions such as ring opening[26] and decarbonylation reaction[27] of anhydrides may take place between the precursors, leading to disordered structures (Supplementary Fig. 2d). To avoid these side reactions, we have developed a templated on-surface synthesis of two types of one-dimensional (1D) covalently bonded structures from different precursor molecules coexisting on the surface. The non-covalent interactions between linear polymers are capable of improving the reaction selectivity as well as the orientation orderliness. In particular, 4, 4″-dibromo-p-terphenyl (DBTP) molecules were codeposited with $Br_4$-PTCDA molecules onto Au(111) surface. At room temperature, owing to the repulsion of bromine substituents, each single $Br_4$-PTCDA molecule has a twisted structure. For most $Br_4$-PTCDA

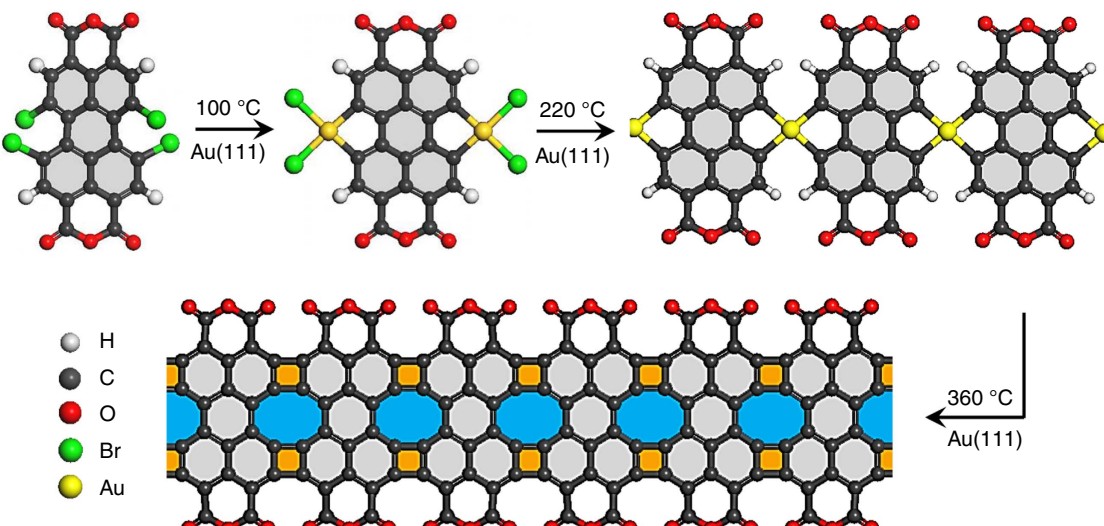

**Figure 1 | Synthetic strategy for graphene-like nanoribbons.** $Br_4$-PTCDA is used as the precursor molecule. At 100 °C, thermal activation induces the C–Br bond cleavage and the formation of PTCDA-$Au_2$-$Br_4$ hybrids on Au(111). At 220 °C, Br atoms are dissociated and linear PTCDA-Au polymers are formed. By further heating to 360 °C, the Au–C bond cleavage and cyclodehydrogenation take place with the formation of graphene-like nanoribbons.

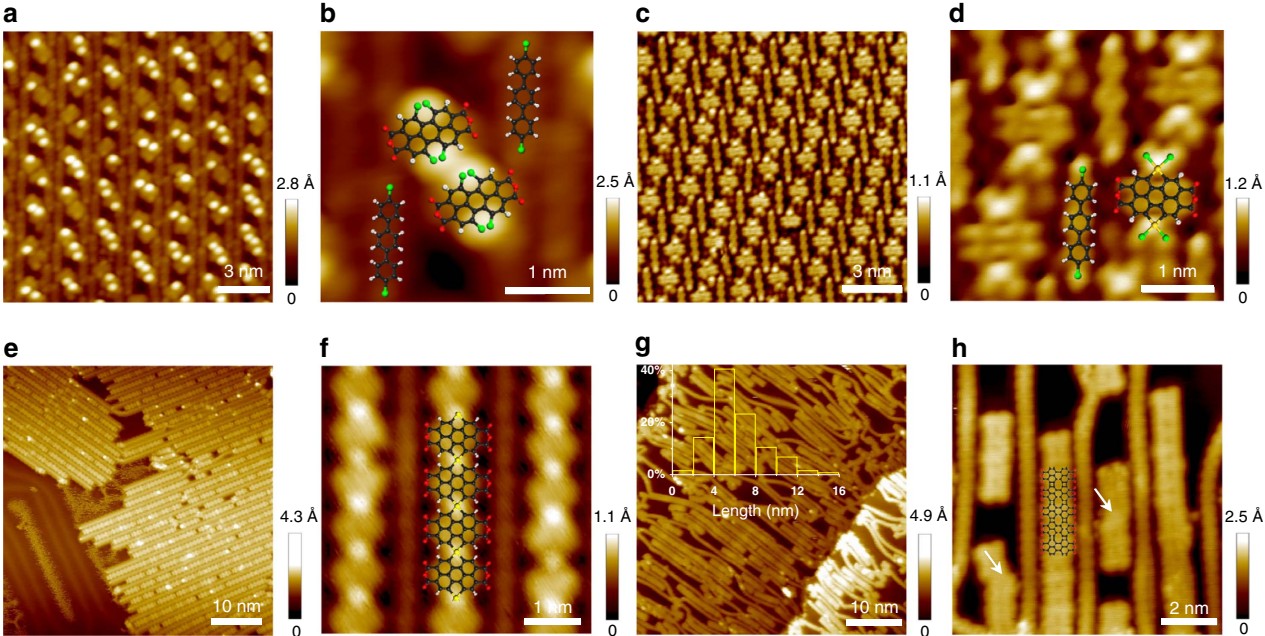

**Figure 2 | Synthesis and characterization of graphene-like nanoribbons.** (**a**) STM image of DBTP and $Br_4$-PTCDA molecules codeposited on Au(111) at RT ($V = -2$ V, $I = 2.2$ nA). (**b**) STM image with overlaid molecular structures. (**c**) STM image of the self-assembled structures of PTCDA-$Au_2$-$Br_4$ hybrids and DBTP molecules on Au (111) prepared at 100 °C ($V = -0.1$ V, $I = 2$ nA). (**d**) High-resolution STM image with partially overlaid molecular model ($V = -0.03$ V, $I = 2.2$ nA). (**e**) PTCDA intermediates colligated with gold atoms into linear polymers between PPP polymers at 220 °C ($V = -2$ V, $I = 0.05$ nA). (**f**) High-resolution STM image with partially overlaid model of the polymer ($V = -0.1$ V, $I = 1.8$ nA). (**g**) STM image of graphene-like nanoribbons comprising four- and eight-membered rings formed after C–Au bond cleavage and cyclodehydrogenation at 360 °C ($V = -1.8$ V, $I = 0.6$ nA). Inset: The ribbon length distribution based on the analysis of a total of 135 ribbons. (**h**) High-resolution STM image with partly overlaid molecular model of the graphene-like nanoribbon ($V = -1.6$ V, $I = 0.3$ nA).

molecules, two bright spots can be clearly observed in Fig. 2a, which correspond to the position of bromine atoms. A well-ordered self-assembled superstructure was formed at 100 °C by the halogen and hydrogen-bond interactions between DBTP molecules and PTCDA-$Au_2$-$Br_4$ hybrids at a ratio of 1:1 (Fig. 2c). The self-assembled bicomponent superstructures were regulated by different growth ratios of the two adsorbed molecules (Supplementary Fig. 4; Supplementary Note 3). By annealing Au(111) surface at 160 °C, the DBTP molecules were linearly polymerized to be poly(para-phenylene) (PPP) polymers after dehalogenation. At the same time, some PTCDA-$Au_2$-$Br_4$ hybrids lost bromines and connected with each other, resulting in PTCDA-Au oligomers coexisting in line with the remaining PTCDA-$Au_2$-$Br_4$ hybrids (Supplementary Fig. 5a). The linear PPP polymers served as the molecular grooves that performed 1D constraint on the PTCDA intermediates. As the substrate temperature was elevated to 220 °C, more Au–Br bonds were cleaved and the PTCDA intermediates diffused unidirectionally along the molecular grooves to form linear PTCDA-Au polymeric chains. The two types of polymers were arranged alternately because of the hydrogen-bond interaction between the phenylene hydrogen and the anhydride oxygen atoms (Fig. 2e).

Further annealing to 360 °C results in nanoribbons showing uniform STM contrasts (Fig. 2g,h), while the protrusions corresponding to Au adatoms in PTCDA-Au polymeric chains are disappeared. The 1D constraint from the PPP polymeric chains in the experiment effectively hampered the interaction of anhydride groups and enabled the formation of flat ribbons stitched together by four- and eight-membered rings. On the other hand, the hydrogen-bond interaction between the graphene-like nanoribbons and PPP polymers hindered the diffusion of nanoribbons to get longer length. The ribbons with 8–12

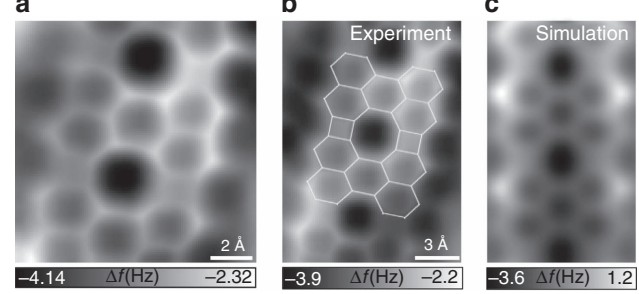

**Figure 3 | Experimental and simulated AFM images of graphene-like nanoribbons.** (**a**) Constant-height nc-AFM frequency shift image of graphene-like nanoribbon resolving four- and eight-membered rings taken with a CO-functionalized tip (oscillation amplitude $A_{OSC} = 1$ Å, $V = 0$ V, $z$ offset $-2$ Å below STM setpoint: $-0.6$ V, 20 pA). (**b**) Constant-height nc-AFM image with partly overlaid ribbon structure. (**c**) Simulated constant-height AFM images. A flexible CO tip ($A_{OSC} = 1$ Å; $k_{tip} = 0.5$ N$m^{-1}$) at the heights of 810 pm is used in the simulation. The online modelling software is provided by Hapala *et al.*[31]

PTCDA monomers were typically observed in our experiment (inset of Fig. 2g), although the longest ribbons with 30 monomers could be obtained. In the high-resolution STM image (Fig. 2h), the periodic contrast variation along the central axis of graphene-like nanoribbons was observed, with dark spots located at the centre of the eight-membered rings. The measured periodic distance between the neighbouring dark spots is $6.3 \pm 0.1$ Å, agreeing well with the distance (6.26 Å) obtained from the optimized structure of graphene-like nanoribbons by DFT calculations. In addition, dislocated cyclodehydrogenation between the PTCDA intermediates resulting in the formation of two fused

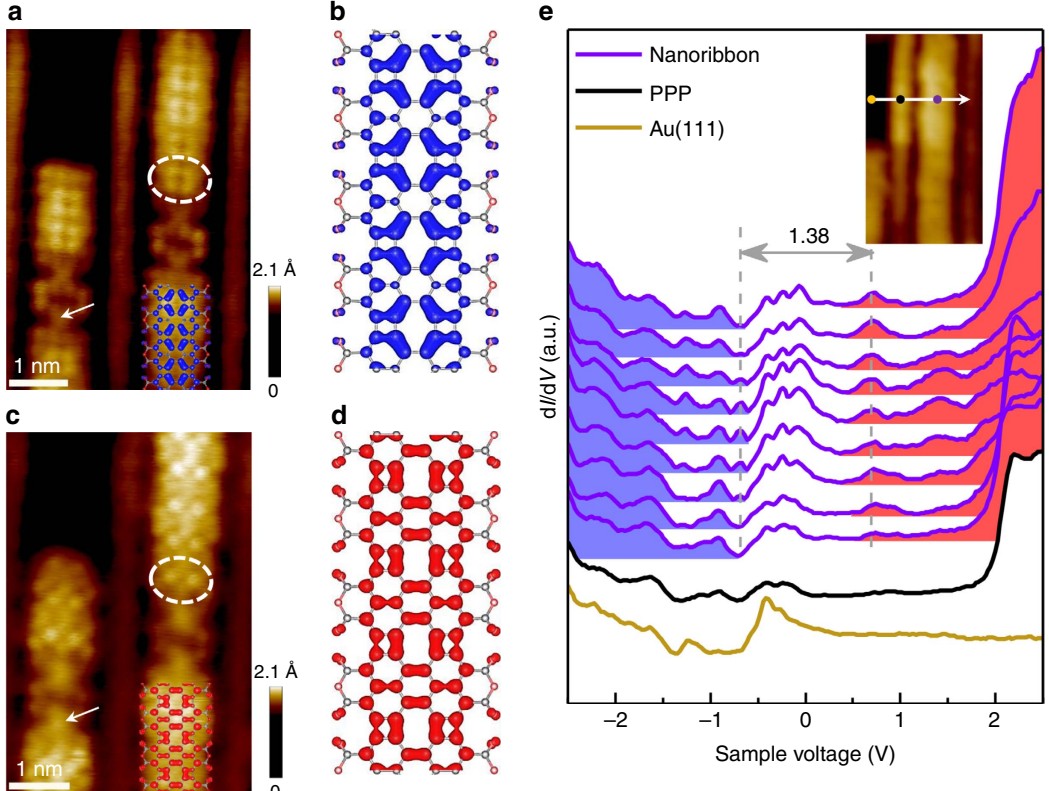

**Figure 4 | HO and LU ribbon states of graphene-like nanoribbons.** (**a,c**) High-resolution STM images of graphene-like nanoribbons with PPP polymers on Au(111) obtained at $V = -0.8$ V (HO ribbon state) and $V = 0.8$ V (LU ribbon state) respectively ($I = 0.6$ nA). (**b,d**) The charge density of the valence band (**b**) and conduction band (**d**) of graphene-like nanoribbons. The contrast in STM images at negative and positive bias voltages well resemble the calculated density distribution. (**e**) Differential conductance (d$I$/d$V$) spectra (purple) taken at different points along a line perpendicular to a graphene-like nanoribbon. The spectra from clean Au(111) (yellow) and PPP polymer (black) are shown as well ($V = -2$ V, $I = 0.6$ nA; modulation voltage $V_{r.m.s.} = 30$ mV).

six-membered rings (arrowed in Fig. 2h) was occasionally observed, similar to a liquid phase reaction reported elsewhere[28]. Compared with the on-surface synthesis without the assistance of linear polymers, the graphene-like nanoribbons obtained by templated on-surface synthesis exhibited a better structural quality (Supplementary Fig. 2d). Besides the PPP polymers, armchair graphene nanoribbons (7-aGNRs)[4] have been used as the molecular templates in our experiment, showing similar constraint effect (Supplementary Fig. 7; Supplementary Note 4).

We performed nc-AFM characterization with a CO-functionalized tip[29,30] to verify the bonding configurations in the nanoribbons. Four- and eight-membered rings formed between adjacent perylene molecules were clearly resolved, as shown in Fig. 3a. The simulated nc-AFM images agree well with the proposed four- and eight-member junction (Fig. 3c)[31,32], confirming that the PTCDA molecules interconnected into ribbon structures with a planar configuration. The observation also indicates that the linear PTCDA-Au polymeric chains underwent demetalation and cyclodehydrogenation at this annealing temperature, which is lower than that of hexagonal carbon rings[4].

**Electronic properties of graphene-like nanoribbons.** PTCDA molecules adsorbed on metal surfaces have been intensively studied and it has been confirmed that PTCDA molecules exhibit rather weak interactions with the Au(111) surfaces[33]. In our experiment, the distribution of highest occupied (HO) and lowest unoccupied (LU) ribbon states are distinguished clearly in the

high-resolution STM images (Fig. 4a,c). To detect the HO and LU states, the bias voltages were set at $-0.8$ and 0.8 V, slightly below the valence band maximum and above the conduction band minimum, respectively. The STM image acquired at $-0.8$ V shows bright feature around the ribbon central axis, with four protrusions located symmetrically at four opposite edges of the eight-membered ring (circled in Fig. 4a). The distribution of HO states is consistent with the characteristic of the calculated result (Fig. 4b). When the bias voltage was set at 0.8 V, different pattern was obtained with broader distribution than the occupied states, as shown in Fig. 4c. There are two bright protrusions symmetrically located around the region of four-membered rings, consistent with the feature of the calculated LU ribbon states (Fig. 4d). In contrast, the parts connected by Au–C bond (arrowed in Fig. 4a,c) in the graphene-like nanoribbons are featureless regardless of positive or negative bias voltage. Our result indicates that the newly formed non-hexagonal carbon rings make a predominant contribution to the frontier electronic states of the ribbons. The coincidence between the high-resolution STM images and the calculated HO and LU states of isolated ribbons implies a rather weak interaction between the graphene-like nanoribbons and the Au(111) surface. In addition, the nanoribbons can be moved on the Au(111) surface by STM manipulation (Supplementary Fig. 6), indicating that there is no strong chemical bonding between the side anhydride groups and the surface.

We have further carried out STS measurements to reveal the energy-dependent local density of states of the graphene-like nanoribbons. Figure 4e shows the differential conductance

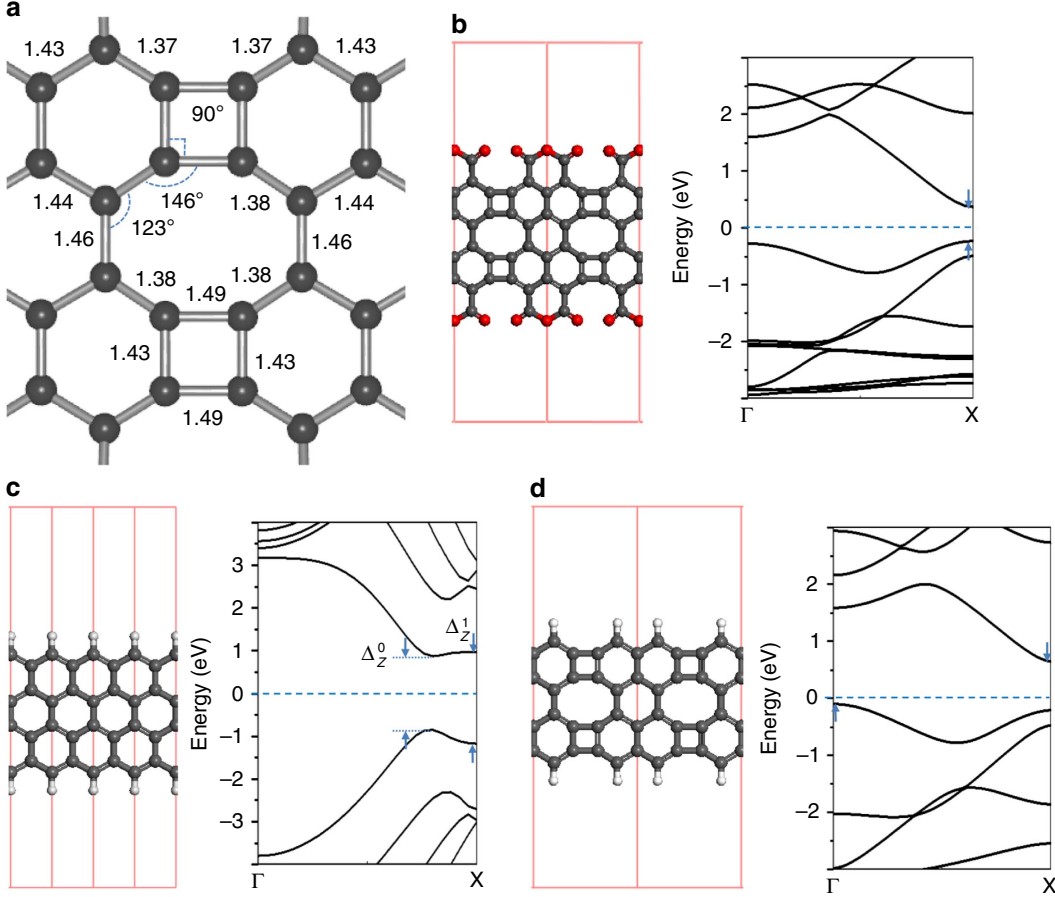

**Figure 5 | Geometric structures and calculated band structures.** (**a**) The optimized geometry of four- and eight-membered carbon rings in graphene-like nanoribbon. The bond lengths shown in **a** are in angstroms. (**b**–**d**) Optimized atomic structures and calculated band structures of the graphene-like nanoribbon, zigzag GNR and graphene-like nanoribbon without anhydride side groups, respectively.

spectra ($dI/dV$) acquired at different positions along a line perpendicular to a graphene-like nanoribbon (inset of Fig. 4e). The characteristic Shockley-type surface state of Au(111) with an onset at $-0.5$ V was obtained on the clean Au(111) surface (yellow curve in Fig. 4e), which contributes to the interface states in the following $dI/dV$ spectra with a broad bump roughly between $-0.5$ and $0.1$ V (Supplementary Fig. 8)[33,34]. The STS curve from a PPP polymer exhibits a prominent peak at 2.2 V corresponding to the lowest unoccupied states (black curve in Fig. 4e)[35]. On the graphene-like nanoribbon, a clear peak appears at $-0.68$ V, which corresponds to the HO states. The intensity of the peak grows as the tip moves from the edge to the ribbon centre, confirming the predominant distribution of the HO states around the central axis (Fig. 4b). On the positive bias side, a peak at 0.70 V was observed, which is assigned to the LU states of the ribbon. No obvious intensity variation of the LU peak was found, showing a broader distribution of LU states on the ribbon. From our STS result, an electronic band gap of $\Delta = 1.38$ eV is derived for the graphene-like nanoribbon with four- and eight-membered carbon rings.

**DFT calculations.** As theoretically predicted, the non-hexagonal carbon rings change not only the density of $\pi$-electron locally, but also the electronic structures and magnetic properties of GNRs[36–38]. To investigate the influence of four- and eight-membered carbon rings, we have carried out DFT calculations to achieve further insight into the effect of four- and eight-membered

rings on the geometric structures and electronic properties of graphene-like nanoribbons. The planar configuration determined by the merged four- and eight-membered rings enables the optimized geometric structure, which led to a delocalized electronic state in our nanoribbons (Fig. 5a). The C–C bonds at the non-hexagonal rings are shortened or lengthened in a range from 1.38 to 1.49 Å and the C–C–C angles are changed in a range from 90° to 146°, because of the missing of hexagonal symmetry[39]. The geometric reconfiguration will alter the local overlap of $P_z$-orbitals in the vicinity of the non-hexagonal rings and accordingly affect the delocalized electronic properties significantly. As shown in Fig. 5b, the calculated band structure of the graphene-like nanoribbon has a direct gap of 0.6 eV, which is underestimated in comparison to our experimental result (1.38 eV). It has been reported that the Heyd–Scuseria–Ernzerhof (HSE06) functional neglects the enhancement of the self-interaction energy at the surface, resulting in the underestimation of the band gap[40,41]. Nevertheless, the essential features of the band structure can be captured by the DFT calculation. Considering a similar GNR with zigzag shaped edges (Fig. 5c), the spin orientation between the two zigzag edges is antiparallel and the antiferromagnetic coupling between the two edges opens a band gap with $\Delta_z^0 = 1.72$ eV and $\Delta_z^1 = 2.15$ eV. However, the spin-polarized edge states are quenched when four- and eight-membered rings are embedded (Fig. 5d). We believe that the spin quenching results from the local rehybridization of $\sigma$ and $\pi$-orbitals. In addition, embedding four- and eight-membered rings in the zigzag GNRs turns the direct energy

gap into indirect and narrower gap (Fig. 5d). According to our calculation, similar band gap narrowing takes place in armchair GNRs by embedding four- and eight-membered rings (Supplementary Fig. 9; Supplementary Note 5).

## Discussion

We successfully fabricated a new type of graphene-like nanoribbons comprising non-hexagonal rings on Au(111) surfaces through a step-like reaction using two precursor molecules coadsorbed on the surfaces. The templated on-surface synthesis is assisted by linear molecular polymers, which were employed to achieve 1D orientation constraint to direct the formation of nanoribbons. Periodically embedded four- and eight-membered rings were identified by nc-AFM characterization. The high-resolution STM images and DFT calculations revealed the predominant distribution of HO and LU states around the eight- and four-membered rings, respectively. The introduction of four- and eight-membered carbon rings different from those with hexagonal geometry characteristic of graphitic structures allows the fine tuning of the band gaps of the ribbons and quench the spin-polarized edge states existing in zigzag GNRs. Undoubtedly, introducing non-hexagonal carbon rings can be an effective engineering approach to modulate the electronic properties of carbon-based nanostructures for achieving desired functionalities.

## Methods

**Experimental measurement.** Single crystalline Au (111) surfaces were cleaned by cycles of $Ar^+$ sputtering and annealing under ultrahigh vacuum (base vacuum $1 \times 10^{-10}$ mbar). $Br_4$-PTCDA molecules were synthesized following a reported method[42]. DBTP and $Br_4$-PTCDA molecules were sublimed from a quartz crucible at sublimation temperatures of 423 and 528 K, respectively, onto the substrate held at room temperature. Samples can be heated by a direct current tungsten filament located on the back side of the sample holder. The sample temperature was measured with a thermocouple. After a succession of stepwise heating processes, the sample was finally heated to 360 °C for 10 min, resulting in the formation of graphene-like nanoribbons comprising four- and eight-membered rings. STM measurements were performed on an Omicron low-temperature STM operated at 78 K. An electrochemically etched tungsten tip was used for topographic and spectroscopic measurements. The STM images were taken in the constant-current mode and the voltages refer to the bias on samples with respect to the tip. The $dI/dV$ spectra were acquired by a lock-in amplifier while the sample bias was modulated by a 553 Hz, 30 mV (r.m.s.) sinusoidal signal under open-feedback conditions. The tip state was checked via the appearance of the characteristic Shockley-type surface state on clean Au(111) surfaces. The nc-AFM measurement was carried out at LHe temperature in constant-height frequency modulation mode with a CO-functionalized tip (resonance frequency $f_0 \approx 40.7$ kHz, oscillation amplitude $A \approx 100$ pm, quanlity factor $Q \approx 5.6 \times 10^4$).

**Theoretical calculation.** Spin-polarized DFT calculations were performed using the periodic plane-wave basis Vienna *ab-initio* Simulation Package code[43,44]. For geometry optimizations and electronic structure calculations, the Perdew–Burke–Ernzerhof functional and the Heyd–Scuseria–Ernzerhof (HSE06) hybrid functional[45] were applied. The valence–core interactions are described using the projector augmented wave method[46]. The plane-wave energy cutoff used for all calculations was 400 eV. The convergence criterion for the forces of structure relaxations is 0.01 eV $Å^{-1}$. A supercell arrangement was used with a 15 Å vacuum layer to avoid spurious interactions between the nanoribbons and periodic images. The electronic structure calculations were performed using a k-point grid of $5 \times 1 \times 1$, including the $\Gamma$-point.

**Data availability.** The data that support the findings of this study are available from the corresponding author upon reasonable request.

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

## Acknowledgements

This work was financially supported by NSFC (Project Nos 11374374, 11574403, 21425310) and the computation part of the work was supported by National Supercomputer Center in Guangzhou.

## Author contributions

Meizhuang Liu and D.Z. conceived the research, designed the experiments and prepared the manuscript. Meizhuang Liu, L.S., Y.H. and Z.C. performed the STM experiments. X.Q., Mengxi Liu, J.P., Z.Z. and S.L. performed the nc-AFM measurements. T.L. and J.W. conducted the synthesis of precursor molecules. Meizhuang Liu conducted the theoretical calculations. Y.Z. helped with the theoretical calculations. D.Z. supervised the work.

## Additional information

**Competing interests:** The authors declare no competing financial interests.

**Publisher's note**: 

