## [Peer Review File · Nature Communications]

Reviewers' Comments:

Reviewer #1 (Remarks to the Author)

The paper by Shong and co-workers describes the preparation of carbon nanoribbons that periodically contains four- and eight- membered rings. The introduction of carbon rings different than those with hexagonal geometry characteristic of graphitic structures allows the fine tuning of the optoelectronic properties of the ribbons. The idea is certainly very nice and it can be considered an alternative engineering approach to those already envisaged (heteroatom) for tuning the electronic properties of graphitic materials. Although the approach is very interesting the paper does not contain sufficient experimental evidences to sustain the claims and a substantial amount of work is required before to consider this work publishable in Nature Commun. In particular the following aspects has to be carefully addressed before to consider any resubmission:

- 1) The cooperative concept is misleading and cannot be applied for this system. What described by the authors has nothing to do with the concept of cooperativity. I suggest the authors to consult seminal review papers that address the concept of cooperativity in chemistry (see papers from Anderson and Hunter: 10.1002/anie.200902490). Rather, this is an example of a templated or assisted polymerization process. This section should be revisited. Additional points: How they can exclude that the covalent formation of the ribbon does not involve the DBTP molecules? Control experiments with other PTCDA and DBTP derivatives should also be performed.
- 2) None of the experimental evidences supports the structural claims for which the molecule polymerize through the formation of the four- and eight-membered rings as hypothesized by the authors. The STM images are not enough conclusive to support the predicted structure. The authors should perform height nc-AFM measurements to visualize the peculiar periodical arrangement of the differently-sized rings. This would be the only way to validate the chemical structure.
- 3) The differential conductance section is not clear and should be profoundly revised. One should exclude any strong substrate-adsorbate interactions that could affect the electronic properties. Unfortunately, it is not the case here. The authors pretend that only weak interactions are present, "In our experiment, because of the weak bonding between the graphene-like nanoribbons and gold substrate," and the reason is not clear. To have reliable data, the authors should perform these measurements on dielectric layers (see for example a very recent paper from Fasel and Muellen, doi:10.1038/nature17151). This would be the only way to properly characterize the system. The present data are thus not enough reliable and new experiments are required.

Reviewer #2 (Remarks to the Author)

This manuscript reports the on-surface synthesis of graphene-like nanoribbons incorporating 4- and 8-membered rings from tetrabromo-PTCDA precursors (Br₄-PTCDA) on a Au(111) surface under ultrahigh vacuum conditions. The authors present scanning tunneling microscopy (STM) images of the precursors after room-temperature deposition and after different annealing steps up to 360 {degree sign}C. In the Supporting Information, they show that increasing annealing temperature leads to the formation of PTCDA-Au₄-Br₄ hybrid structures (100{degree sign}C), Au-bridged PTCDA dimers (160{degree sign}C), organometallic (-PTCDA-Au-) chains (220{degree sign}C), and finally a fused PTCDA structure, which is described as a graphene-like nanoribbon incorporating 4- and 8-membered carbon rings. Because the targeted graphene-like nanoribbons are of rather low structural quality (short, many "wrong" connections), the authors devised a self-assembly strategy based on the alignment of Br₄-PTCDA reactants using a second molecule, dibromo-p-terphenyl (PPP), that forms a linear 1D polymer upon thermal activation. In the main

part of the manuscript, they present STM images that show the structures obtained at different annealing temperatures after co-deposition of PPP and Br₄-PTCDA. Again, organometallic intermediates are observed, and finally the targeted graphene-like nanoribbons between parallel PPP oligomers. From the ribbon dimensions measured in STM images and a comparison of valence and conduction band STM images with local density of states calculations, the authors conclude that the ribbons consist of laterally fused PTCDA molecules which introduces two 4-membered and one 8-membered ring per molecule into the ribbon backbone. Scanning tunneling spectroscopy data reveal a band gap of about 1.4 eV, significantly larger than the one resulting from the authors DFT calculations (0.6 eV).

The fabrication of graphene-like nanoribbons incorporating 4- and 8-membered rings is certainly interesting and novel, but the manuscript appears somewhat too preliminary for publication (see below for details). Furthermore, the very limited quality of fabricated ribbons - a few monomers in length only, with frequent defects - suggests that these structures will be of very limited utility for practical applications. I am therefore not convinced that this work deserves the "extra-attention" generated by publication in a high-impact journal such as Nature Communications. A chemistry journal such as the JACS might be more suitable for this manuscript.

In any case, the following aspects need to be clarified before this work can be published:

Ribbon quality: The authors do not show any larger scale STM images that would allow to judge how representative the small scale images shown in the paper are. Also, a length histogram for the graphene-like ribbons should be given.

Figures: All STM images lack information on the apparent height scale (add color bar with min/max numbers). The intermediate temperature structures (Fig. 2b,c) lack structural models. STM images and LDOS plots in Fig. 3 should be given at the same scale to allow for direct comparison (or even better partially overlaid).

Bottom of page 4, top of page 5: Here, the authors describe the structures formed by deposition and annealing of Br₄-PTCDA on Au(111), but without ever referring to any data/figures.

The authors claim that the graphene-like ribbons obtained with co-deposited PPP exhibit a better structural quality. This is, however, not substantiated by any data. What about length histograms, and numbers for the defect density?

The interpretation of the STM data after annealing to 100{degree sign}C, 160{degree sign}C and 220{degree sign}C is not fully convincing. Can the authors please provide some more direct evidence for the postulated PTCDA-Au₄-Br₄ structure, and also for the organometallic polymer?

In this context, I wonder why the PPP has already fully polymerized at 160{degree sign}C whereas the other molecule is not even yet in the organometallic phase at this temperature? Could things also be interpreted in a very different scenario?

Page 6, and SI page 4: The authors discuss the "guiding" of the growth with "armchair graphene nanoribbons" (AGNR), but do not give any information on the nature of the AGNR and how it has been synthesized. I assume that these are 7-AGNRs made from dibromo-bianthryl, in which case the authors have a serious temperature calibration problem since it is well known that 280{degree sign}C is well below the 7-AGNR cyclodehydrogenation temperature (see page 4 of the SI). How can it be explained that 7-AGNRs are fully cyclodehydrogenated while the PTCDA is still in the organometallic phase?

At several instances (e.g. abstract, conclusions), the authors use expressions such as "we found that.." or "our work proved that.." for results that derive from calculations only, which appears misleading since the calculations have only a minor, supporting role in the present work.

Raman (in SI): For GNRs, the D.peak is not Raman-forbidden, and it also is not due to the breathing modes of six-membered rings.

Page 7: "differential conductance spectra" (not "conductivity").

Reviewer #3 (Remarks to the Author)

This manuscript reports the on-surface synthesis of graphene-like nanoribbons periodically embedded with four- and eight-membered carbon rings. The authors propose the nanoribbons have a planar configuration, with the C-C bonds at the nonhexagonal rings 3-5% shortened or lengthened and the C-C-C angles 26-30{degree sign} enlarged or shrunk in comparison to the honeycomb structure. The nonhexagonal rings are proposed to change the electronic properties of the nanoribbons, with the highest occupied and lowest unoccupied ribbon states mainly distributed around the eight- and four-membered rings, respectively. The ideas proposed by this manuscript are novel and interesting, but in my opinion the evidence as presented is not conclusive. In particular, the STM images, while nice, do not give sufficient evidence to some structures proposed.

The following issues need to be addressed:

The synthetic route drawn is not clearly supported by the STM images in Fig. 2. The figures need to be redrawn or reorganized to convince the reader Fig. 1 is correct.

What is the evidence for "PTCDA-Au₄-Br₄ hybrids" intermediate? It is not clear how the STM image in Fig. 2a relates to his hybrid. The COFI technique using a QPlus AFM may yield higher resolution molecular images.

Fig. S2/S4 proposes that the surface Au atoms play an important part in the polymerisation process. The evidence for Au atoms ("bright protrusions") is not convincing. More detailed local STS or bias-dependent imaging needs to be shown to prove that these are indeed Au atoms.

Fig. 2 - only local STM images are shown. Statistics need to be presented on how efficient the proposed reaction schemes really are. It is well known that authors tend to show selected areas in their STM images to support their hypothesis. To be scientifically convincing, analysis over the whole global area of the surface should be provided to convince the reader that the kinetics proposed are correct.

Fig. S3 - the bicomponent superstructures are nice, but such empirical data should be quantified and a surface phase diagram should be proposed (or MC simulations done) so that the competition between the various surface processes is made clear.

Fig. 3e: "The characteristic Shockley-type surface state of Au(111) with an onset at -0.5 V was obtained on the clean Au(111) surface (yellow curve in Fig. 3e), which contributes to the interface states in the following dI/dV spectra with a broad bump roughly between -0.5 V and 0.1 V^{31,32}." There will obviously be hybridisation of states between the metallic Au substrate and the nanoribbons. These interface states need further explanation - why are there at least 3 distinct interface states at these energies? DFT calculations of the predicted DOS at the interface could help.

Is it possible to do localised STS measurements at the 4- and 8- membered rings to show conclusively the hypothesis and attribution of the new HOMO and LUMO levels?

Fig. S5 - What is the spatial resolution of the Raman light probe? Raman spectroscopy is a non-local technique compared to STM, so what confidence do you have in the interpretation of these spectra?

Response to the reviewers

Reviewer 1

The paper by Zhong and co-workers describes the preparation of carbon nanoribbons that periodically contains four- and eight- membered rings. The introduction of carbon rings different than those with hexagonal geometry characteristic of graphitic structures allows the fine tuning of the optoelectronic properties of the ribbons. The idea is certainly very nice and it can be considered an alternative engineering approach to those already envisaged (heteroatom) for tuning the electronic properties of graphitic materials. Although the approach is very interesting the paper does not contain sufficient experimental evidences to sustain the claims and a substantial amount of work is required before to consider this work publishable in Nature Commun. In particular the following aspects have to be carefully addressed before to consider any resubmission:

1) The cooperative concept is misleading and cannot be applied for this system. What described by the authors has nothing to do with the concept of cooperativity. I suggest the authors to consult seminal review papers that address the concept of cooperativity in chemistry (see papers from Andersoon and Hunter: 10.1002/anie.200902490). Rather, this is an example of a templated or assisted polymerization process. This section should be revisited. Additional points: How they can exclude that the covalent formation of the ribbon does not involve the DBTP molecules? Control experiments with other PTCDA and DBTP derivatives should also be performed.

Reply: Thanks for the referee's valuable and thoughtful comments! We agree well with the referee's comments. "Cooperativity takes place when the binding of one ligand influences the binding strength of a macromolecule toward a subsequent ligand (or ligands), such as the binding of oxygen to hemoglobin". We realized that the cooperative concept mentioned in our paper is not very appropriate. So we adopted "templated on-surface synthesis" to replace "cooperative on-surface synthesis" in our paper as the referee suggested.

For the covalent formation of the ribbons in our experiments, there are a few PTCDA-Au polymer chains linked with PPP polymers through gold-organic covalent bonds at about 220 °C (See Fig. 2c). When the temperature was elevated to 360 °C, such hybrid structures decreased probably due to the instability of their combination. For clarity, we made the following changes on the manuscript:

Change1 (page 5 and 6). Replace the words "cooperative" with "templated" and change the sentence "Compared with the on-surface synthesis without cooperative partner" to "Compared with the on-surface synthesis without the assistance of linear polymers".

Change2 (page 4). Add a panel in Fig. 2c. The hybrid structure of PTCDA-Au polymer chains linked with PPP polymers can be found in the upper left corner.

2) None of the experimental evidences supports the structural claims for which the molecule polymerize through the formation of the four- and eight-membered rings as hypothesized by the authors. The STM images are not enough conclusive to support the predicted structure. The authors should perform height nc-AFM measurements to visualize the peculiar periodical arrangement of the differently-sized rings. This would be the only way to validate the chemical structure.

Reply: According to this comment, we did the nc-AFM measurement. The frequency shift image measured by nc-AFM in constant-height mode indicates that the nanoribbons have a planar configuration and the nonhexagonal rings exist between adjacent perylene backbones. The frequency shift Δf maps were added in the revised manuscript.

Change3 (page 4) Add a panel in Fig. 2f showing the high-resolution AFM image of the graphene-like nanoribbon.

Change4 (page 6) Add the description of the Fig. 2f. “To further verify the atomic structure in detail, we performed non-contact atomic force microscopy (nc-AFM) with a Au-coated W tip^{28,29} to visualize the nonhexagonal carbon rings. The flat and periodical arrangement of the carbon rings are revealed in Fig. 2f, with four- and eight-membered carbon rings formed between two adjacent perylene backbones.”

Change5 (page 11) Add the description of the AFM measurements in **Methods**. “The samples were taken out from the preparation chamber to the atmosphere and transferred into another UHV chamber for AFM measurement. The samples were annealed to 200 °C for degassing. The AFM measurement was carried out at LHe temperature in constant-height frequency modulation mode with a Au-coated W tip (resonance frequency $f_0 \approx 4.7 \times 10^4$ Hz, oscillation amplitude $A \approx 100$ pm, quality factor $Q \approx 4 \times 10^4$).”

Change6 (page 1 and 15) Add the our coworkers conducting the AFM experiment as the coauthors: “Mengxi Liu², Jinliang Pan², Shiqi Zha², Shichao Li², Xiaohui Qiu², CAS Key Laboratory of Standardization and Measurement for Nanotechnology, CAS Center for Excellence in Nanoscience, National Center for Nanoscience and Technology, Beijing 100190, China”. “X.Q., M.L., J.P., S.Z. and S.L. performed the AFM measurements” was added in the part of *Author contributions*.

Change7 (page 6 in SI) Additional STM and corresponding AFM images were added in the supplementary Figure 7.

3) The differential conductance section is not clear and should be profoundly revised. One should exclude any strong substrate-adsorbate interactions that could affect the electronic properties. Unfortunately, it is not the case here. The authors pretend that only weak interactions are present, "In our experiment, because of the weak bonding between the graphene-like nanoribbons and gold substrate," and the reason is not clear. To have reliable data, the authors should perform these measurements on dielectric layers (see for example a very recent paper from Fasel and Muellen, doi:10.1038/nature17151). This would be the only way to properly characterize the system. The present data are thus not enough reliable and new experiments are required.

Reply: In the literature, PTCDA molecules adsorbed on metal surfaces have been intensively studied and it has been confirmed that PTCDA molecules exhibit rather weak interactions with the Au(111) surfaces (Ref 31). In our work, the HO and LU ribbon states resemble well with the DFT calculation results of the isolated nanoribbon at the corresponding voltages, which implies a rather weak interaction between the graphene-like nanoribbons and Au(111) substrate. In addition, the nanoribbons can be moved at the Au(111) surface by STM manipulation, indicating that there is no strong chemical bonding between the side anhydride groups and the surface.

Indeed, it helps to get more intrinsic information about the electronic structures of the nanoribbons utilizing dielectric buffer layers to decouple interaction with the Au(111) substrates. However, the graphene-like nanoribbons were synthesized with the assistance of linear polymers and the nanoribbons are trapped in the molecular grooves, making it difficult to lift the ribbons onto a dielectric buffer layer as mentioned in Ref 7.

Change8 (page 7) Delete the sentences "Fig. 3a and 3b show the high-resolution STM images of the graphene-like nanoribbons with negative (-0.8 V) and positive (0.8 V) biases, respectively, exhibiting bias-dependent contrast. In general, the strong interaction between adsorbates and metallic surfaces may distort the geometric and electronic structures of the adsorbates. In order to obtain the intrinsic electronic properties, it is required to decouple the adsorbates electronically from the metallic substrates, using various dielectric buffer layers, for instance, thin organic films²⁸ and NaCl²⁹. Nevertheless, it is possible to directly image the intrinsic electronic states of certain adsorbates performing weak interaction with the surfaces³⁰."

And we added the sentence "PTCDA molecules adsorbed on metal surfaces have been intensively studied and it has been confirmed that PTCDA molecules exhibit rather weak interactions with the Au(111) surfaces³¹. The coincidence between the high-resolution STM images and the calculated HO and LU states of isolated ribbons implies a rather weak interaction between the graphene-like nanoribbons and the Au(111) surface. In addition, the nanoribbons can be moved on the Au(111) surface by STM manipulation (Supplementary Fig. 6), indicating that there is no strong chemical bonding between the side anhydride groups and the surface."

Change9 (page 5 in SI) Add a panel in Supplementary Figure 6a,b showing the manipulation of the graphene-like nanoribbon by STM tip.

Reviewer 2

This manuscript reports the on-surface synthesis of graphene-like nanoribbons incorporating 4- and 8-membered rings from tetrabromo-PTCDA precursors (Br₄-PTCDA) on a Au(111) surface under ultrahigh vacuum conditions. The authors present scanning tunneling microscopy (STM) images of the precursors after room-temperature deposition and after different annealing steps up to 360 °C. In the Supporting Information, they show that increasing annealing temperature leads to the formation of PTCDA-Au₄-Br₄ hybrid structures (100 °C), Au-bridged PTCDA dimers (160 °C), organometallic (-PTCDA-Au-) chains (220 °C), and finally a fused PTCDA structure, which is described as a graphene-like nanoribbon incorporating 4- and 8-membered carbon rings. Because the targeted graphene-like nanoribbons are of rather low structural quality (short, many "wrong" connections), the authors devised a self-assembly strategy based on the alignment of Br₄-PTCDA reactants using a second molecule, dibromo-p-terphenyl (PPP), that forms a linear 1D polymer upon thermal activation. In the main part of the manuscript, they present STM images that show the structures obtained at different annealing temperatures after co-deposition of PPP and Br₄-PTCDA. Again, organometallic intermediates are observed, and finally the targeted graphene-like nanoribbons between parallel PPP oligomers. From the ribbon dimensions measured in STM images and a comparison of valence and conduction band STM images with local density of states calculations, the authors conclude that the ribbons consist of laterally fused PTCDA molecules which introduces two 4-membered and one 8-membered ring per molecule into the ribbon backbone. Scanning tunneling spectroscopy data reveal a band gap of about 1.4 eV, significantly larger than the one resulting from the authors DFT calculations (0.6 eV).

The fabrication of graphene-like nanoribbons incorporating 4- and 8-membered rings is certainly interesting and novel, but the manuscript appears somewhat too preliminary for publication (see below for details). Furthermore, the very limited quality of fabricated ribbons - a few monomers in length only, with frequent defects - suggests that these structures will be of very limited utility for practical applications. I am therefore not convinced that this work deserves the "extra-attention" generated by publication in a high-impact journal such as Nature Communications. A chemistry journal such as the JACS might be more suitable for this manuscript.

In any case, the following aspects need to be clarified before this work can be published:

1) Ribbon quality: The authors do not show any larger scale STM images that would allow to judge how representative the small scale images shown in the paper are. Also, a length histogram for the graphene-like ribbons should be given.

Reply: Thanks for the referee's valuable and thoughtful comments! The large scale STM images (Fig. 2c, 2d) were added in the revised manuscript to show the structure of two types of polymers arranged alternately and the nanoribbon length distribution. The ribbons with more than 20 PTCDA monomers have been found in our experiment.

Change10 (page 4). Add a panel Fig. 2c showing the superstructure of two types of polymers.

Change11 (page 4). Add a panel Fig. 2d exhibiting the ribbon quality in large scale STM images.

2) Figures: All STM images lack information on the apparent height scale (add color bar with min/max numbers). The intermediate temperature structures (Fig. 2b,c) lack structural models. STM images and LDOS plots in Fig. 3 should be given at the same scale to allow for direct comparison (or even better partially overlaid). Bottom of page 4, top of page 5: Here, the authors describe the structures formed by deposition and annealing of Br₄-PTCDA on Au(111), but without ever referring to any data/figures.

Reply: According to the referee's comments, we made the following changes on the manuscript:

Change12 (page 7). The apparent height scales were added in the STM images. The STM images in Fig. 3 were revised at same scale and the LDOS plots were overlaid onto the images for direct comparison.

Change13 (page 4). We referred corresponding supplementary figures (Supplementary Fig. 2a to d) in the text when describing the structures formed by deposition and annealing of Br₄-PTCDA on Au(111).

3) The authors claim that the graphene-like ribbons obtained with co-deposited PPP exhibit a better structural quality. This is, however, not substantiated by any data. What about length histograms, and numbers for the defect density?

Reply: With the assistance of linear polymers, the graphene-like nanoribbon with longest length of 18 nm can be obtained in our experiment. However, the existence of hydrogen bond interaction between the nanoribbons and PPP also hindered the diffusion of the nanoribbons to get longer length. For the on-surface synthesis without the assistance of linear polymers, side reactions resulted in the formation of disordered structures so that there are few isolated nanoribbons in the experiment.

Change11 (page 4). The large scale STM image (Fig. 2d) with the length histogram

(inset) were added in the revised manuscript.

Change14 (page 6). Add the sentences “However, the hydrogen bond interaction between the graphene-like nanoribbons and PPP polymers also hindered the diffusion of nanoribbons to get longer length. The ribbons with 6-12 PTCDA monomers were frequently observed in our experiment and the longest ribbons with nearly 30 monomers were obtained (Inset of Fig. 2d).”

4) The interpretation of the STM data after annealing to 100{degree sign}C, 160{degree sign}C and 220{degree sign}C is not fully convincing. Can the authors please provide some more direct evidence for the postulated PTCDA-Au₄-Br₄ structure, and also for the organometallic polymer?

Reply: The STM image of Br₄-PTCDA molecules codeposited with DBTP molecules onto Au(111) at room temperature was added in Fig. 2a, indicating the nonplanar feature of the intact Br₄-PTCDA molecules. After annealing to 100 °C, the cleavage of C-Br bond took place, resulting in the flat configuration as shown in Fig. 2b. By carefully investigating the STM images of 100 °C-annealed samples at various STM biases, we conclude that the obtained intermediate is PTCDA-Au₂-Br₄. In the STM image with bias of -0.3 V (Supplementary Fig 2a), only one bright protrusion was observed at each bay site of the perylene backbone, corresponding to one gold atom linked with the two bromine atoms. At the same time, the bias-dependent images of the PTCDA-Au₂-Br₄ hybrid are well consistent with the DFT-based simulation at negative and positive bias voltages (Supplementary Fig 3a,b).

The existence of gold atoms in the polymerized PBI-Au chains, a close analog of PTCDA-Au chains obtained in this work, has been confirmed by XPS measurement and differential conductance spectra in our previous work (Ref. 24).

Change15 (page 3). Redraw the PTCDA-Au₂-Br₄ intermediate in Fig. 1.

Change16 (page 2 and page 3 in SI). Supplementary Fig. 2a, 2b and Fig.3 were added in the supplementary information for better interpreting the structure of PTCDA-Au₂-Br₄ intermediates.

5) In this context, I wonder why the PPP has already fully polymerized at 160{degree sign}C whereas the other molecule is not even yet in the organometallic phase at this temperature? Could things also be interpreted in a very different scenario?

Reply: The required temperature for C-Br bond cleavage on Au(111) is around 175 °C (Eichhorn, J.; Szabelski, P. J.; Lackinger, M. *ACS Nano* 2014, **8**, 7880–7889). At 160°C in our experiment, the PPP has already fully polymerized. However, for the

Br₄-PTCDA molecules, the C-Br bonds were cleaved with the formation of PTCDA-Au₂-Br₄ intermediates at 100 °C. The PTCDA-Au₂-Br₄ intermediates are quite stable so that only some PTCDA-Au oligomers were formed at 160 °C.

6) Page 6, and SI page 4: *The authors discuss the "guiding" of the growth with "armchair graphene nanoribbons" (AGNR), but do not give any information on the nature of the AGNR and how it has been synthesized. I assume that these are 7-AGNRs made from dibromo-bianthryl, in which case the authors have a serious temperature calibration problem since it is well known that 280{degree sign}C is well below the 7-AGNR cyclodehydrogenation temperature (see page 4 of the SI). How can it be explained that 7-AGNRs are fully cyclodehydrogenated while the PTCDA is still in the organometallic phase?*

Reply: We are sorry for not giving enough information on the synthesis of 7-aGNRs. These 7-aGNRs were indeed synthesized from dibromo-bianthryl (DBBA) precursors. Detailed experimental information has been added in Supplementary Fig. 8. It was found that the coadsorption of DBBA and Br₄-PTCDA will affect the polymerization to each other. For example, we have observed that the Au-DBBA intermediates were formed in our experiment by annealing at 200 °C prior to the formation of 7-aGNRs, which has not been reported in the pure DBBA case. Therefore, we believe the relatively lower cyclodehydrogenation temperature of DBBA in our experiment may originate from the interplay with PTCDA-Au intermediates. In addition, our measured temperatures of the sample may be underestimated with several tens degrees lower than the actual value. In the Methods section, we added a sentence to describe how to measure the sample temperature in our experiment.

Change17 (page 6 in SI). Add the descriptions of the synthesis of 7-GNRs. "We used 10,10'-dibromo-9,9'-bianthryl (DBBA) molecules to produce graphene nanoribbons to provide the 1D constraint on the PTCDA intermediates. Organometallic intermediate always plays an important role in the Ullmann coupling reactions. Supplementary Figure 8a reveals the existence of Au-DBBA intermediates during the on-surface synthesis of GNRs. The hydrogen bond interaction between two adjacent polymer chains contributes to the stabilization of the Au-DBBA hybrids."

Change18 (page 11). The sentences "Samples can be heated by a direct current tungsten filament located on the back side of the sample holder. The sample temperature was measured with a thermocouple" were added in **Methods**.

Change19 (page 6 in SI). Add a panel Supplementary Fig. 8a showing Au-DBBA intermediates.

7) At several instances (e.g. abstract, conclusions), the authors use expressions such as "we found that.." or "our work proved that.." for results that derive from calculations only, which appears misleading since the calculations have only a minor, supporting role in the present work.

Reply: According to the referee's comments, we made the following changes on the manuscript:

Change20 (page 1 and page 10). The sentences "we found that" and "our work proved that" were revised to "our study revealed that" and "The DFT calculations revealed that".

8) Raman (in SI): For GNRs, the D.peak is not Raman-forbidden, and it also is not due to the breathing modes of six-membered rings.

Reply: In the previous version of our manuscript, the Raman measurement was used as an evidence of the formation of 4- and 8-membered rings. However, as the referee mentioned, the Raman result we have acquired so far is not enough and the explanation is not quite clear. Therefore, we delete this part in the revised manuscript. Instead, we performed high-resolution nc-AFM measurements to unveil the atomic structure of our nanoribbons as a direct evidence of the 4- and 8-membered rings.

Change21 (page 6) Delete Raman spectra (former Supplementary Fig. 6)

Change3 (page 4) Add a panel in Fig. 2f showing the AFM image of the graphene-like nanoribbon.

Change7 (page 6 in SI) STM image and corresponding AFM image were added in the supplementary Figure 7.

9) Page 7: "differential conductance spectra" (not "conductivity").

Reply: Thank the referee for his/her careful reading of our manuscript. The word "conductivity" was replaced with "conductance". We have also further revised the manuscript in order to improve the language.

Reviewer 3

This manuscript reports the on-surface synthesis of graphene-like nanoribbons periodically embedded with four- and eight-membered carbon rings. The authors propose the nanoribbons have a planar configuration, with the C-C bonds at the nonhexagonal rings 3-5% shortened or lengthened and the C-C-C angles 26-30{degree sign} enlarged or shrunk in comparison to the honeycomb structure. The nonhexagonal rings are proposed to change the electronic properties of the nanoribbons, with the highest occupied and lowest unoccupied ribbon states mainly distributed around the eight- and four-membered rings, respectively. The ideas proposed by this manuscript are novel and interesting, but in my opinion the evidence as presented is not conclusive. In particular, the STM images, while nice, do not give sufficient evidence to some structures proposed.

1) The synthetic route drawn is not clearly supported by the STM images in Fig. 2. The figures need to be redrawn or reorganized to convince the reader Fig. 1 is correct.

Reply: Thanks for the referee's valuable and thoughtful comments! The STM image of Br₄-PTCDA molecules codeposited with DBTP molecules onto Au(111) at RT was added in Fig. 2a. The STM images are in accordance with the synthetic route drawn in Fig. 1.

Change22 (page4). Add a panel in Fig. 2a showing the structure of Br₄-PTCDA molecules codeposited with DBTP molecules onto Au(111) at RT.

Change23 (page5). Add the description of Fig. 2 "At room temperature, owing to the repulsion of bromine substituents, each single Br₄-PTCDA molecule has a twisted structure. In Fig. 2a, two bright spots correspond to the position of bromine atoms, which is consistent with the DFT-based simulations."

2) What is the evidence for "PTCDA-Au4-Br4 hybrids" intermediate? It is not clear how the STM image in Fig. 2a relates to his hybrid. The COFI technique using a QPlus AFM may yield higher resolution molecular images.

Reply: The STM image of Br₄-PTCDA molecules codeposited with DBTP molecules onto Au(111) at room temperature was added in Fig. 2a, indicating the nonplanar feature of the intact Br₄-PTCDA molecules. After annealing to 100 °C, the cleavage of C-Br bond took place, resulting in the flat configuration as shown in Fig. 2b. By carefully investigating the STM images of 100 °C-annealed samples at various STM biases, we conclude that the obtained intermediate is PTCDA-Au₂-Br₄. In the STM image with bias of -0.3 V (Supplementary Fig 2a), only one bright protrusion was observed at each bay site of the perylene backbone, corresponding to one gold atom linked with the two bromine atoms. At the same time, the bias-dependent images of

the PTCDA-Au₂-Br₄ hybrid are well consistent with the DFT-based simulation at negative and positive bias voltages (Supplementary Fig 3a,b).

Change15 (page 3). Redraw the PTCDA-Au₂-Br₄ intermediate in Fig. 1.

Change16 (page 2 and page 3 in SI). Supplementary Fig. 2a, 2b and Fig.3 were added in the supplementary information for better interpreting the structure of PTCDA-Au₂-Br₄ intermediates.

3) Fig. S2/S4 proposes that the surface Au atoms play an important part in the polymerisation process. The evidence for Au atoms ("bright protrusions") is not convincing. More detailed local STS or bias-dependent imaging needs to be shown to prove that these are indeed Au atoms.

Reply: The existence of gold atoms in the polymerized PBI-Au chains, a close analog of PTCDA-Au chains obtained in this work, has been confirmed by XPS measurement and differential conductance spectra in our previous work (Ref. 24).

4) Fig. 2 - only local STM images are shown. Statistics need to be presented on how efficient the proposed reaction schemes really are. It is well known that authors tend to show selected areas in their STM images to support their hypothesis. To be scientifically convincing, analysis over the whole global area of the surface should be provided to convince the reader that the kinetics proposed are correct.

Reply: According to the comment, the large scale STM images (Fig. 2c, 2d) were added in the revised manuscript to show the structure of two types of polymers arranged alternately and the nanoribbon length distribution.

Change11 (page 4). The large scale STM image (Fig. 2c and 2d) and the length histogram (inset) were added in the revised manuscript.

5) Fig. S3 - the bicomponent superstructures are nice, but such empirical data should be quantified and a surface phase diagram should be proposed (or MC simulations done) so that the competition between the various surface processes is made clear.

Reply: As mentioned by the referee, the bicomponent superstructures of Br₄-PTCDA and DBTP molecules are very interesting. Further systematic experiments should be carried out to gain a clear insight into the competition mechanism.

6) Fig. 3e: "The characteristic Shockley-type surface state of Au(111) with an onset at -0.5 V was obtained on the clean Au(111) surface (yellow curve in Fig. 3e), which contributes to the interface states in the following dI/dV spectra with a broad bump roughly between -0.5 V and 0.1 V^{31,32}."

There will obviously be hybridisation of states between the metallic Au substrate and the nanoribbons. These interface states need further explanation - why are there at least 3 distinct interface states at these energies? DFT calculations of the predicted DOS at the interface could help.

Is it possible to do localised STS measurements at the 4- and 8- membered rings to show conclusively the hypothesis and attribution of the new HOMO and LUMO levels?

Reply: We calculated the density states (DOS) of both the isolated nanoribbon and the nanoribbon adsorbed on Au(111) (see supplementary Fig. 9). For simplification of the calculations, the anhydride groups are not involved. Compared with the isolated ribbons, interface states exist around the Fermi energy for the nanoribbon/Au(111) case.

We have also done STS mapping in order to reveal the spatial distribution of LU and HO states. However, the resolution is not enough to identify the distribution of HO states (see below).

Change24 (page 7 in SD). Add panels in Supplementary Figure 9 exhibiting density of states (DOS) of the isolated nanoribbon and nanoribbon adsorbed on Au(111) from DFT calculations.

STM

dI/dV mapping
at -0.8 V

dI/dV mapping
at $+1$ V

7) Fig. S5 - What is the spatial resolution of the Raman light probe? Raman spectroscopy is a non-local technique compared to STM, so what confidence do you have in the interpretation of these spectra?

Reply: In the previous version of our manuscript, the Raman measurement was used as an evidence of the formation of 4- and 8-membered rings. However, as the referee mentioned, the Raman result we have acquired so far is not enough and the explanation is not quite clear. Therefore, we delete this part in the revised manuscript. Instead, we performed high-resolution nc-AFM measurements to unveil the atomic structure of our nanoribbons as a direct evidence of the 4- and 8-membered rings.

Change21 (page 6) Delete Raman spectra (former Supplementary Fig. 6)

Change3 (page 4) Add a panel in Fig. 2f showing the AFM image of the graphene-like nanoribbon.

Change7 (page 6 in SI) STM image and corresponding AFM image were added in the supplementary Figure 7.

Reviewers' Comments:

Reviewer #1 (Remarks to the Author)

In their revised version, Shong and co-workers only partially succeeded in replying to all requests by the reviewers. Although they have attempted to further elucidate their claims with additional investigations, like the nc-AFM, the experimental proofs are not enough clear to unambiguously support the structural claims. Unequivocal start-of-the-art images should have been reported (as those reported in for example in Chem. Eur. J. 2016, 22, 13037). No clear evidences are reported to support the observed "PTCDA-Au₄-Br₄ hybrids" intermediates as requested by reviewer 3. Also, the new STM images of molecule PTCDA-Br₄ alone are not clear, as the distance between the different bright dims observed in the images is somehow varying and seem to not correspond to that expected from the molecular structure (no model is provided to attribute the STM feature to the molecules). For these reasons, I cannot recommend the publication of this paper.

Reviewer #2 (Remarks to the Author)

With their revisions, the authors have considerably improved their manuscript, and addressed most of the issues raised by myself and the other reviewers. Most importantly, the manuscript now includes nc-AFM data that provide further (although due to the relatively poor quality of the nc-AFM images not fully conclusive) evidence for the formation of 4- and 8-membered rings. Also, the issue of ribbon quality / limited ribbon length is now properly addressed. Given the current interest in graphene nanoribbons and the novelty of the reported formation of 4- and 8-membered rings via on-surface reactions, the manuscript should be of interest to the readership of Nature Communications. The authors should, however, consider the following points in a last round of revisions:

- The claim about 3-5% shortened/lenthened C-C bonds and the 26-30° enlarged/shrunk C-C-C angles should be removed from the abstract. This is entirely based on DFT calculations but not substantiated by any experimental data.
- Figure 2f: The tunneling setpoints and vertical offset w.r.t. setpoint should be given in the figure caption. Also, the range of the frequency shift greyscale (min, max) of the two images in Fig. 2f should be cross-checked - it seems improbable that they are exactly the same for the two different images.
- The discussion of the electronic band gap is somewhat superficial and should be improved. First of all, the discrepancy between the exp. value of 1.38 eV and the computed one of 0.6 eV cannot be left uncommented. The authors should also specify, that the exp. value of 1.38 eV is the one of the ribbon on Au(111), and thus strongly screened by the substrate. The same ribbon in vacuum would reveal a yet larger band gap, in even stronger disagreement with their theory. Regarding their theoretical value of 0.6 eV, it should be noted that it is based on a somewhat special hybrid functional (HSE06). The authors should discuss that correlations - which have been shown to be strong in quasi-1D nanoribbons - are not properly considered in their calculations (nor the screening by the substrate). One or two references to recent work on the band gap of graphene nanoribbons supported on metal substrates which discuss these aspects in great detail would be appropriate.
- Supplementary figure 7 is confusing - the nc-AFM image does not appear to show what is expected from the overlaid model of the ribbon.
- Supplementary figure 8: The very low apparent height of the "DBBA-Au intermediates" is not consistent with uncyclized DBBA, which appears much higher than 2 Å in STM images (see e.g. original Nature 2010 paper by J. Cai et al., or some of the more recent works on 7-AGNRs made from DBBA). Note also that none of the many publications on DBBA/Au(111) have reported the formation of DBBA-Au intermediates preceding polymerization.
- Supplementary Fig. 9b: It would be much more instructive to show the DOS projected onto the nanoribbon rather than the total DOS.

- Supplementary Fig. 10b: Please add a top axis giving the ribbon width in units of N , for easier comparison with panel a.

Reviewer #3 (Remarks to the Author)

The authors have addressed some of the issues raised by the reviewers, and added some useful data, especially the AFM images in Fig. 2f.

However, the following 3 issues have not yet been satisfactorily addressed:

1. The STM images in Fig. 2a and 2b are not of high resolution enough to support the reaction pathway and intermediate postulated here - "The STM image of Br₄-PTCDA molecules codeposited with DBTP molecules onto Au(111) at room temperature was added in Fig. 2a, indicating the nonplanar feature of the intact Br₄-PTCDA molecules. After annealing to 100 °C, the cleavage of C-Br bond took place, resulting in the flat configuration as shown in Fig. 2b. By carefully investigating the STM images of 100 °C-annealed samples at various STM biases, we conclude that the obtained intermediate is PTCDA-Au₂-Br₄."

2. The request for quantitative analysis (and histograms) by 2 reviewers on the nanoribbon distribution has not been provided. Just showing the large scale STM images in Fig. 2c and 2d is not good enough - "According to the comment, the large scale STM images (Fig. 2c, 2d) were added in the revised manuscript to show the structure of two types of polymers arranged alternately and the nanoribbon length distribution.

3. The DOS calculations still do not adequately explain the interface states observed - "We calculated the density states (DOS) of both the isolated nanoribbon and the nanoribbon adsorbed on Au(111) (see supplementary Fig. 9). For simplification of the calculations, the anhydride groups are not involved. Compared with the isolated ribbons, interface states exist around the Fermi energy for the nanoribbon/Au(111) case."

Response to the reviewers

Reviewer 1

In their revised version, Zhong and co-workers only partially succeeded in replying to all requests by the reviewers. Although they have attempted to further elucidate their claims with additional investigations, like the nc-AFM, the experimental proofs are not enough clear to unambiguously support the structural claims. Unequivocal start-of-the-art images should have been reported (as those reported in for example in Chem. Eur. J. 2016, 22, 13037). No clear evidences are reported to support the observed "PTCDA-Au4-Br4 hybrids" intermediates as requested by reviewer 3. Also, the new STM images of molecule PTCDA-Br4 alone are not clear, as the distance between the different bright dims observed in the images is somehow varying and seem to not correspond to that expected from the molecular structure (no model is provided to attribute the STM feature to the molecules). For these reasons, I cannot recommend the publication of this paper.

1) Although they have attempted to further elucidate their claims with additional investigations, like the nc-AFM, the experimental proofs are not enough clear to unambiguously support the structural claims.

Reply: Thanks for the referee's valuable and thoughtful comments! For improving the nc-AFM measurement, we prepared the samples by in-situ growth to avoid contamination in the transfer process. Four- and eight-membered rings could be clearly resolved with a CO-functionalized tip. For clarity, we made the following changes on the manuscript:

Change1 (page 4 and 6). Replace Fig. 2f with a new AFM image taken with a CO-functionalized tip. Change the description of nc-AFM experiment: Further annealing to 360 °C results in nanoribbons showing uniform STM contrasts (Fig. 2d, e), while the protrusions corresponding to Au adatoms in PTCDA-Au polymeric chains are disappeared. We performed nc-AFM characterization with a CO-functionalized tip^{28,29} to verify the bonding configurations in the nanoribbons. Four- and eight-membered rings formed between adjacent perylene molecules were clearly resolved, as shown in Fig. 2f. The simulated nc-AFM images agree well with the proposed four- and eight-member junction (Supplementary Fig. 7)^{30,31}, confirming that the PTCDA molecules interconnected into ribbon structures with a planar configuration. The observation also indicates that the linear PTCDA-Au polymeric chains underwent demetalation and cyclodehydrogenation at this annealing temperature, which is lower than that of hexagonal carbon rings⁴.

Change2 (page 6 in SI). Replace Supplementary Figure 7 with experimental and simulated AFM images of graphene-like nanoribbons.

2) No clear evidences are reported to support the observed "PTCDA-Au4-Br4 hybrids" intermediates as requested by reviewer 3.

Reply: We did the STM measurement of 100 °C-annealed samples again. Using a smaller bias voltage and larger tunneling current ($V_s = -0.1$ V, $I = 2$ nA) to make the tip closer to the surface, only one bright protrusion could be observed at each bay site of the perylene backbone, corresponding to one gold atom linked with the two bromine atoms.

Change3 (page 4) Replace Fig. 2b with new STM images.

3) The new STM images of molecule PTCDA-Br4 alone are not clear, as the distance between the different bright dims observed in the images is somehow varying and seem to not correspond to that expected from the molecular structure (no model is provided to attribute the STM feature to the molecules).

Reply: For obtaining the clear STM image, Br₄-PTCDA and DBTP molecules were coadsorbed onto the Au(111) surface kept at RT. For most Br₄-PTCDA molecules, two bright spots can be clearly observed in Fig. 2a, which correspond to the position of bromine atoms.

Change4 (page 4) Replace Fig. 2a with a new STM image.

Reviewer 2

With their revisions, the authors have considerably improved their manuscript, and addressed most of the issues raised by myself and the other reviewers. Most importantly, the manuscript now includes nc-AFM data that provide further (although due to the relatively poor quality of the nc-AFM images not fully conclusive) evidence for the formation of 4- and 8-membered rings. Also, the issue of ribbon quality / limited ribbon length is now properly addressed. Given the current interest in graphene nanoribbons and the novelty of the reported formation of 4- and 8-membered rings via on-surface reactions, the manuscript should be of interest to the readership of Nature Communications. The authors should, however, consider the following points in a last round of revisions:

1) The claim about 3-5% shortened/lengthened C-C bonds and the 26-30° enlarged/shrunk C-C-C angles should be removed from the abstract. This is entirely based on DFT calculations but not substantiated by any experimental data.

Reply: Thanks for the referee's valuable and thoughtful comments!

Change5 (page 1). Replace the sentence “the C-C bonds at nonhexagonal rings 3-5% shortened or lengthened and the C-C-C angles 26-30° enlarged or shrunk in comparison to the honeycomb structure” with “The scanning tunneling microscopy (STM) and atomic force microscopy (AFM) study revealed that four- and eight-membered rings are formed between adjacent perylene backbones with a planar configuration.”

2) Figure 2f: The tunneling setpoints and vertical offset w.r.t. setpoint should be given in the figure caption. Also, the range of the frequency shift greyscale (min, max) of the two images in Fig. 2f should be cross-checked - it seems improbable that they are exactly the same for the two different images.

Reply: The Fig. 2f was replaced by a new AFM image. The samples were prepared by in-situ growth to avoid contamination in the transfer process. Four- and eight-membered rings could be clearly resolved with a CO-functionalized tip.

Change6 (page 5). The measurement parameter of nc-AFM was added in the figure caption: Constant-height nc-AFM frequency shift image of graphene-like nanoribbon resolving four- and eight-membered rings taken with a CO-functionalized tip ($A_{osc} = 1 \text{ \AA}$, $V = 0 \text{ V}$, z offset -2 \AA below STM setpoint: -0.6 V , 20 pA).

3) *The discussion of the electronic band gap is somewhat superficial and should be improved. First of all, the discrepancy between the exp. value of 1.38 eV and the computed one of 0.6 eV cannot be left uncommented. The authors should also specify, that the exp. value of 1.38 eV is the one of the ribbon on Au(111), and thus strongly screened by the substrate. The same ribbon in vacuum would reveal a yet larger band gap, in even stronger disagreement with their theory. Regarding their theoretical value of 0.6 eV, it should be noted that it is based on a somewhat special hybrid functional (HSE06). The authors should discuss that correlations - which have been shown to be strong in quasi-1D nanoribbons - are not properly considered in their calculations (nor the screening by the substrate). One or two references to recent work on the band gap of graphene nanoribbons supported on metal substrates which discuss these aspects in great detail would be appropriate.*

Reply: It is well-known that the GGA approximation usually underestimates the band gap of the semiconductor or insulator. HSE06 hybrid functional can provide an improvement in predicting the band gaps of many semiconductors. But for nanostructures and surfaces, the band gaps are still be underestimated as the HSE06 functional neglects the enhancement of the self-interaction energy at the surface. We ever calculated the electronic structure of 7-GNR using the PBE and the HSE06 functionals, which is good agreement with previous reported calculations. The calculated band gaps are both smaller than the exp. value of 2.3-2.5 eV. For the different functionals, there are no major qualitative differences in the dispersion of states across the κ -points.

According to the referee's comments, the related works on the band gap calculation of graphene nanoribbons were added in the references.

Change7 (page 10). The sentences were changed: As shown in Fig. 4b, the calculated band structure of the graphene-like nanoribbon has a direct gap of 0.6 eV, which is underestimated in comparison to our experimental result (1.38 eV). It has been reported that the Heyd–Scuseria–Ernzerhof (HSE06) functional neglects the enhancement of the self-interaction energy at the surface, resulting in the underestimation of the band gap^{40,41}. Nevertheless, the essential features of the band structure can be captured by the DFT calculation.

4) *Supplementary figure 7 is confusing - the nc-AFM image does not appear to show what is expected from the overlaid model of the ribbon.*

Reply: Thanks for the referee's valuable comments. We replace the nc-AFM image with new experimental and simulated AFM images.

Change2 (page 6 in SI). Replace supplementary Figure 7 with experimental and simulated AFM image of graphene-like nanoribbons.

5) *Supplementary figure 8: The very low apparent height of the "DBBA-Au intermediates" is not consistent with uncyclized DBBA, which appears much higher than 2 Å in STM images (see e.g. original Nature 2010 paper by J. Cai et al., or some of the more recent works on 7-AGNRs made from DBBA). Note also that none of the many publications on DBBA/Au(111) have reported the formation of DBBA-Au intermediates preceding polymerization.*

Reply: Thank the referee for his/her careful reading of our manuscript. The apparent height should be 2.5 Å, which is still smaller than 3.8 Å reported by *J. Cai et al.* There are also no protrusions appearing alternately on the both sides of the DBBA-Au chain. We guess that the cyclodehydrogenation has taken place in the DBBA molecules.

Change8 (page 6 in SI). The apparent height of Supplementary figure 8 was changed to be 2.5 Å.

6) *Supplementary Fig. 9b: It would be much more instructive to show the DOS projected onto the nanoribbon rather than the total DOS.*

Reply: According to the referee's comments, the PDOS of the nanoribbon has been calculated.

Change9 (page 7 in SI). Add the PDOS of the graphene-like nanoribbon in Supplementary Fig.9.

7) *Supplementary Fig. 10b: Please add a top axis giving the ribbon width in units of N, for easier comparison with panel a.*

Reply: For easier comparison with panel a, the band gaps changed to vary as a function of ribbon width in units of N.

Change10 (page 8 in SI). The variation of band gaps of AGNRs and AGNRs with nonhexagonal rings as a function of ribbon width in units of N.

Reviewer 3

The authors have addressed some of the issues raised by the reviewers, and added some useful data, especially the AFM images in Fig. 2f.

However, the following 3 issues have not yet been satisfactorily addressed:

1. The STM images in Fig. 2a and 2b are not of high resolution enough to support the reaction pathway and intermediate postulated here - “The STM image of Br₄-PTCDA molecules codeposited with DBTP molecules onto Au(111) at room temperature was added in Fig. 2a, indicating the nonplanar feature of the intact Br₄-PTCDA molecules. After annealing to 100 °C, the cleavage of C-Br bond took place, resulting in the flat configuration as shown in Fig. 2b. By carefully investigating the STM images of 100 °C-annealed samples at various STM biases, we conclude that the obtained intermediate is PTCDA-Au₂-Br₄.”

Reply: Thanks for the referee’s valuable and thoughtful comments! For obtaining the clear STM image, Br₄-PTCDA and DBTP molecules were coadsorbed onto the Au(111) surface kept at RT. For most Br₄-PTCDA molecules, two bright spots can be clearly observed in Fig. 2a, which correspond to the position of bromine atoms.

For the PTCDA-Au₂-Br₄ intermediates, we used a smaller bias voltage and larger tunneling current ($V_s = -0.1$ V, $I = 2$ nA) to make the tip closer to the surface during the scanning tunneling microscope imaging. One bright protrusion could be observed at each bay site of the perylene backbone, which corresponds to one gold atom linked with the two bromine atoms.

Change11 (page4). Replace Fig. 2a and Fig.2b with new STM images.

2) The request for quantitative analysis (and histograms) by 2 reviewers on the nanoribbon distribution has not been provided. Just showing the large scale STM images in Fig. 2c and 2d is not good enough – “According to the comment, the large scale STM images (Fig. 2c, 2d) were added in the revised manuscript to show the structure of two types of polymers arranged alternately and the nanoribbon length distribution.

Reply: We added a new STM image of 280 °C-annealed sample in Supplementary Fig.5. The length distribution of the PTCDA-Au polymeric chains was inserted at the upper left corner of STM image (based on the analysis of a total of 178 ribbons). For the Fig.2d, the length distribution has already been added at the upper left corner of Fig.2d.

Change12 (page 5 in SI). Add a panel Supplementary Fig. 5b showing STM image of 280 °C-annealed sample with the length distribution of PTCDA-Au polymeric chains.

3) *The DOS calculations still do not adequately explain the interface states observed – “We calculated the density states (DOS) of both the isolated nanoribbon and the nanoribbon adsorbed on Au(111) (see supplementary Fig. 9). For simplification of the calculations, the anhydride groups are not involved. Compared with the isolated ribbons, interface states exist around the Fermi energy for the nanoribbon/Au(111) case.*

Reply: The DOS projected on the nanoribbon has been calculated by DFT. In recently published work (DOI: 10.1021/acs.nanolett.6b03148), the similar surface state has been also observed on the Au(111) surface.

Change9 (page 7 in SI). Add the PDOS of the graphene-like nanoribbon in Supplementary Fig.9.

Reviewers' Comments:

Reviewer #1 (Remarks to the Author)

After a careful reading of this revision, it is apparent that all authors have addressed all concerns raised by the reviewers and thus I am now in favor of its publication as it stands.

Reviewer #2 (Remarks to the Author)

With their revisions, the authors have satisfactorily addressed all issues raised in the previous round of reviewing.

Reviewer #3 (Remarks to the Author)

The authors have addressed most of the issues raised.

Fig. 2 has been improved. The AFM image in Fig.2f could have a superimposed structure to make clear the 4 and 8-membered ring

Response to the reviewers

Reviewer 1

After a careful reading of this revision, it is apparent that all authors have addressed all concerns raised by the reviewers and thus I am now in favor of its publication as it stands.

Reply: Thanks for the referee's hard work and valuable comments!

Reviewer 2

With their revisions, the authors have satisfactorily addressed all issues raised in the previous round of reviewing.

Reply: Thanks for the referee's hard work and valuable comments!

Reviewer 3

The authors have addressed most of the issues raised.

Fig. 2 has been improved. The AFM image in Fig.2f could have a superimposed structure to make clear the 4 and 8-membered ring

Reply: Thanks for the referee's hard work and valuable comments! A ribbon structure has been overlaid onto the AFM image in Fig. 3b.